# Microstructure Evolution and Strengthening Mechanism of Galvanized Steel/Mg Alloy Joint Obtained by Ultrasonic Vibration-Assisted Welding Process

**DOI:** 10.3390/ma14071674

**Published:** 2021-03-29

**Authors:** Fangzhou Yang, Bing Liu

**Affiliations:** College of Materials Science and Engineering, Chongqing University of Arts and Sciences, Chongqing 402160, China; 20050001@cqwu.edu.cn

**Keywords:** ultrasonic vibration, galvanized steel, magnesium, welding, microstructure, mechanical properties

## Abstract

A novel ultrasonic vibration-assisted welding (UVAW) process was used to achieve reliable joining of galvanized steel and Mg alloy. The effects of the UVAW technique on the microstructure and mechanical properties of galvanized steel/Mg alloy weldment were studied in detail. The introduction of ultrasonic vibration can ameliorate the wetting of welds and eliminate porosity defects. A refined microstructure of the fusion welding zone with an average grain size of 39 ± 1.7 µm was obtained and attributed to cavitation and acoustic streaming caused by the UVAW process. The grain refinement led to an increase in the microhardness and joining strength of the galvanized steel/Mg alloy weldment. Under the ultrasonic power of 0.9 kW and a current of 65 A, the maximum joining strength of the ultrasound-treated galvanized steel/Mg alloy joint was 251 ± 4.1 MPa, which was a 14.6% increase over the joint without ultrasonic treatment.

## 1. Introduction

With increasing demand from consumers and the environment for high fuel efficiency vehicles, as well as for the reduction of carbon dioxide emissions from vehicles, nonferrous metals, as a good substitute for traditional metal alloys, have attracted much attention [1,2,3]. Specifically, nonferrous alloys based on aluminum (Al) [4], magnesium (Mg) [5,6], and titanium (Ti) [7] with high specific strength, superior processability, and easy recycling have been developed and used in automobile manufacturing and aerospace fields. Over the past decade, Mg alloys have attracted considerable attention in the automotive industry because of their potential to reduce weight to achieve better fuel economy [8,9]. However, in modern vehicles designed with a variety of materials, the connection of Mg alloy to the underlying substructure (steel) may be a challenge.

The current study was mainly focused on the fabrication of steel/Mg composite components by adding intermediate metal layers [10,11]. Xu et al. [12] studied the effects of an aluminized zinc coating on the tensile strength and microscopic interface characteristics of steel/Mg weldment. The results showed that during the friction stir welding process, the interface wettability of the steel/Mg was significantly enhanced by the Zn coating, and the metallurgical connection between the steel and Mg alloy was promoted by Al_5_Fe_2_ phase in the Zn coating on the surface of the steel side matrix. Zhao et al. [13] investigated the laser welding-brazing of Ni-coated Q235A-steel and AZ31B-Mg alloy. According to Zhao, there is a strong mutual attraction between Al and Ni atoms in the weld. The maximum tensile/shear fracture load was close to 230MPa and the weld efficiency reached 88.5% relative to the AZ31B-Mg alloy.

However, the way to ameliorate the performance of steel/Mg joints through a metal intermediate layer has reached a bottleneck. For example, it was difficult to eliminate the structural defects, such as coarse grains and pores, on the Mg alloy side of the joint with a traditional welding process. In comparison with other welding techniques, ultrasonic vibration-aided welding is an emerging welding technology with advantages of high efficiency and low energy consumption [14,15]. Liu et al. [16] studied the influence of ultrasonic vibration on welding the residual stress, micro-hardness, and microscopic characteristics of Q345 steel weld. The results showed that the conventional weld of Q345 steel has a distinct micro-hardness gradient and columnar grains, while ultrasonic impact-treated weld was characterized by a uniform micro-hardness and equiaxed grains [16]. Hu et al. [17] reported the influences of ultrasonic treatment on elevated-temperature tensile strength and the microstructure evolution of hot-extruded Mg–6Al–0.8Zn–2.0Sm wrought Mg alloy. The experimental results revealed that the morphology of Al_2_Sm phase was fine granular after ultrasonic vibration treatment. In addition, the maximum pressure induced by the collapse of cavitation bubbles was much greater than the shear strength of Al_2_Sm phase, which caused the rough petal-like Al_2_Sm phase to split into fine polygonal granules [17]. Wang et al. [18] introduced ultrasonic vibration into an underwater flux-cored arc welding process. According to Wang, ultrasonic vibration can affect the size and morphology of austenite grains during the solidification of the weld pool and ultimately promote the formation of a large number of fine ferrite structures in the subsequent solid phase transformation of the weld metal. In addition, ultrasonic vibration made the columnar structure of the welded metal more refined, achieving a good balance of high tensile strength and impact toughness.

The present study of ultrasonic vibration-assisted welding mainly focused on the application of vibration to the entire weldment during the welding process. Although the above process had achieved the results of refining the microstructure and improving properties, this process consumed more energy and had potential hazards of welding deformation. If ultrasonic vibration can be accurately applied to the local area of the weld pool, it is expected to greatly improve the welding quality. Therefore, the present study attempted to introduce ultrasonic vibration into the liquid metal of molten pool through filler wire, aiming to improve the microstructure and performance while minimizing the adverse effects of vibration on the entire weldment. The evolution of joint appearances, the weld microstructure, and mechanical performances were systematically investigated.

## 2. Materials and Methods

### 2.1. Selected Materials

Galvanized steel (Hangtai, Shandong, China) and AZ31 Mg alloy (Hongya, Henan, China) were used as parent metals. Table 1 shows the main chemical composition of the parent metals. Galvanized steel belongs to the ferritic and martensitic duplex steel system, which has an average grain size of about 17 ± 1.1 µm. The average thickness of the galvanized layer on the steel surface was about 14 ± 0.6 µm, as shown in Figure 1a. The AZ31 welding stick, with a diameter of 1.2mm, was adopted as filler material. Figure 1b shows the metallographic map of the Mg alloy parent material. It can be seen from the figure that the Mg alloy base material contained a large number of approximate equiaxed crystals with an average grain size of about 19 ± 0.8 µm.

### 2.2. UVAW Process

To ensure the joining quality, the Mg alloy base metal and filler material were firstly burnished using abrasive papers (1000) to remove the surface oxides. All the base metals were then immersed in anhydrous alcohol for about 1 minute to remove surface oil contamination. Finally, all the base metals were dried with cold air and stored in a drying oven. A self-made ultrasonic vibration auxiliary welding system, including an argon tungsten arc welding machine (YC-300WP5HGN, Panasonic, Tangshan, China), a cuboid ultrasonic generator (GDZ-1012M, Longke, Dongguan, China), a conical amplitude transformer, a columnar transducer (DW-20k, Longke, Dongguan, China), and so forth, was used for the welding of the galvanized steel and Mg alloy. The ultrasonic frequency was 20 kHz and the amplitude was 30 µm. Figure 2 illustrates the sketch map of the ultrasonic vibration-assisted welding process. The galvanized steel plate and Mg alloy plate were fixed by a lap joint (both with the dimensions of 70 mm × 50 mm × 2 mm) with an overlapping size of 16 mm, and the welding torch was placed above the edge of the Mg alloy plate. During the UVAW process, the ultrasonic vibration signal was produced by the generator, amplified by the transformer, output by the horn, and finally, acted on the welding wire. In consequence, the liquid weld metal was subjected to horizontal ultrasonic vibration during the UVAW process. Table 2 shows the welding parameters applied in the study.

### 2.3. Joint Characterization

After the UVAW process, the transverse cross-sections of the lap joints were cut from the weldment for metallographic observation. The specimens (12 mm × 8 mm × 4 mm) used for metallurgical observation were treated with sandpaper (600#, 1000#, and 2000#) and polished with distilled water, and then immersed into a picric acid solution (5 mL acetic acid, 5 g picric acid, 5 mL distilled water, and 30 mL alcohol) for 10 s. Microstructural observation of the weld was carried out with an optical microscope (OM, CR20-530HS, Beite, China) and scanning electron microscope (SEM, VAGA 2 LMH, TESCAN, Brno, Czech Republic) in backscattered electron (BSE, VAGA 2 LMH, TESCAN, Brno, Czech Republic) mode. The chemical composition analysis of the weld zone was performed via energy-dispersive X-ray spectroscopy (EDS; VAGA 2 LMH, TESCAN, Brno, Czech Republic). The wetting angle and width of the weld were measured by the tangent method and the metallographic positioning method, respectively. A uniaxial tensile test was conducted using a universal tensile testing machine (AG-X, SHIMADZU, Kyoto, Japan), which was operated at room temperature with a constant stretching speed of 2.2 mm/min. The sampling point and outline dimensions (GB/T 2651-89) of the tensile samples are shown in Figure 3. For each set of welding parameters, the mean value of five tensile test results was adopted. The fracture samples were cleaned by ultrasonic with acetone and the fracture surfaces were observed by SEM in secondary electron (SE) mode to determine the failure characteristics. The microhardness distributions of the weld joint were evaluated along the horizontal directions with a 50 g load and a 10 s holding time (MH-5L).

## 3. Results and Discussion

### 3.1. Weld Cross-Sections

Figure 4 presents the typical cross-sectional area of weld obtained with and without the UVAW process. Under the heat input of the welding torch, the AZ31B Mg alloy parent metal and filler material melted to form a weld zone (WZ), while a brazing zone (BZ) appeared between the galvanized steel plate and the weld zone. As shown in Figure 4a, micropores were formed in the weld area of the weldment, which were mainly attributed to the hydrogen evolution characteristics of Mg alloy and the rapid solidification of the welding process. There is no doubt that hydrogen pores, as a welding defect, will deteriorate the mechanical properties of the weldment and make it unreliable. It is interesting to note that the pores of the weld area disappeared with the aid of proper ultrasonic vibration, as presented in Figure 4b. This is mainly due to the fact that after the formation of hydrogen bubbles in the molten bath, the application of high-frequency vibration is beneficial for bubbles to float up and leave the molten pool, thereby avoiding the porosity formation in the weld zone. In addition to the elimination of welding defects, Figure 5 indicates that the ultrasonic vibration process also affects the wetting and seam width of the joint. The ultrasonic vibration introduced into the molten bath by the welding stick had a stirring effect on the liquid metal, which eventually promoted the wetting and spreading of the molten pool on the upper surface of the steel substrate and increased the weld width.

### 3.2. Weld Microstructure

Figure 6a presents the typical cross-sectional profile of galvanized steel/Mg alloy weldment, which reveals dual characteristics of fusion welding and brazing. Namely, the welding wire, the AZ31B parent plate, and the galvanized layer melt to form the fusion welding zone (Zone B), while the high melting point of the steel substrate and the molten metal (Mg alloy parent plate, welding wire, and galvanized layer) formed the brazing area (Zone A).

Figure 6b–d show the corresponding BSE micrographs of the brazing area labeled as A, which were welded with and without the assistance of ultrasonic vibration. Under a current of 55 A, a uniform and continuous flake-like reaction layer, with a thickness of about 2 µm, was formed in the interfacial area, as presented in Figure 6b. No crack defects were observed due to the direct contact between the steel substrate and the interfacial reaction layer. Under a current of 65 A, a corrugated reaction layer was formed in the interface area, as shown in Figure 6c. It is obvious that the increase in heat input changed the morphology of the reaction layer and increased its thickness. This was mainly due to the enhanced heat input that can cause the upward volatilization trend of the molten galvanized layer in the molten pool, which caused the undulation of the brazing layer. On the other hand, the increase in the welding heat input increased the bath temperature and enhanced the atomic activity in the interface area, which was conducive to the interface reaction. Therefore, the morphology of the reaction layer in the brazing area was changed, as described above. Figure 6d shows the BSE micrograph of the brazing zone of the weld with the assistance of the UVAW process. By comparing Figure 6c with Figure 6d, it can be seen that the application of the ultrasonic treatment had a certain impact on the morphology of the reaction layer. The main manifestation was that the wave-like morphology of the reaction layer was weakened and the wave tip disappeared. During the ultrasonic vibration aid welding process, the application of the UVAW process played a role in stirring the molten bath, which made the temperature and composition of the molten pool uniform, and finally, inhibited the formation of a wave-like reaction layer. To identify the phase components of the interfacial reaction layer in the brazing area, EDS was carried out and the results show that phases P1 and P2 mainly contained 71.6 wt.% pct Zn, 28.4 wt.% pct Mg and 70.8 wt.% pct Zn, 29.2 wt.% pct Mg, respectively. It was suggested that both phase P1 and P2 were MgZn, which indicates that the UVAW process only changed the morphology of the reaction layer but had no influence on the phase composition.

Figure 7 shows the corresponding metallographic microstructure of the fusion welding areas labeled as B (as marked in Figure 6a), which were obtained with and without the assistance of the UVAW process. As presented in Figure 7a, significant microstructure coarsening occurred in the fusion welding area of the conventional tungsten inert gas (TIG) welded joint. The average crystallite dimension of the fusion welding area was about 63 ± 5.3 µm, which was 3.3 times that of the AZ31B Mg alloy parent metal. However, it is worth noting that the coarse structure of the fusion zone was significantly improved as ultrasonic vibration was applied to the dissimilar metal welding process of the galvanized steel and Mg alloy, as shown in Figure 7b. The influence of ultrasonic power on the crystallite dimension of the weld zone is illustrated in Figure 7c. It is obvious that with the increase of ultrasonic power, the influence of the UVAW process on the crystal refinement of the weld area increased. For instance, the refined microstructure of the fusion welding zone, with an average grain size of 39 ± 1.7 µm, was obtained under the ultrasonic power of 0.9 kW. However, after the ultrasonic power exceeded 1.2 kW, continuing to increase the ultra-high power no longer had a significant impact on the microstructure refinement of the weld zone. Xu et al. [19] also reported a similar pattern of microstructure evolution.

Figure 8 illustrates the refinement mechanism of the UVAW process on the microstructure of the weld zone. The literature reveals that the ultrasonic cavitation and acoustic streaming caused by ultrasonic treatment account for the refinement of coarsening grain in the fusion welding zone. As presented in Figure 8a, a large number of cavitation bubbles were generated in the negative half cycle of the ultrasonic sine wave. Then, the cavitation bubbles absorbed the heat around them and grew up rapidly, resulting in supercooling in the vicinity and thus, forming a mass of nuclei, as illustrated in Figure 8b. When the cavitation bubble grew to the threshold value, it burst and produced a shock wave of gigapascal, thus breaking the growing dendrites, as shown in Figure 8c. Therefore, both undercooled nucleation and broken dendrite can improve the nucleation rate and refine the microstructure of the fusion zone. In addition, during the process of ultrasonic vibration propagation in the weld zone, the sound pressure gradient caused by viscosity attenuation made the liquid metal flow. Under the action of acoustic streaming, the supercooled nucleation points and broken dendrites were evenly transported to various areas of the molten pool. In addition, the acoustic flow accelerated the uniform distribution of elements and alleviated the enrichment of solute at the solidification front, which eventually promoted the spheroidization of α-Mg grains, as shown in Figure 8d.

### 3.3. Weld Microhardness

In order to facilitate the comparison of the hardness of the ultrasonic-treated joint and the untreated joint, indentations were carried out along the straight lines in the joint cross-section, as shown in Figure 9. As shown in the figure, ultrasonic treatment had a noteworthy effect on the hardness distribution of the weld in both horizontal and vertical directions. Under proper ultrasonic treatment, the average microhardness of the welding zone was increased obviously. The improvement of the microhardness is mainly attributed to the grain refinement caused by UVAW. According to Xu, the Tabor empirical formula reveals that the relationship between weld microhardness and grain size can be expressed by the following formula [19]:(1)$$H=C\left(\sigma_{0}+kd^{-1/2}\right)$$
where *H* is the weld microhardness value, *C* is a constant related to the material, *σ*_0_ is the friction stress of the crystal lattice, *k* is the gradient constant, and *d* is the average grain diameter. The above formula indicates that the microhardness of the weld is inversely proportional to the average crystallite dimension. In consequence, the grain refinement caused by UVAW is a key factor in improving the welding strength and microhardness of the weld.

### 3.4. Tensile Strength and Fracture Characteristic

Figure 10 illustrates the tensile strength of the Mg alloy base metal, untreated and ultrasound-treated galvanized steel/Mg alloy joints. For the galvanized steel/Mg alloy joints without the UVAW process, the increase in welding heat input promoted the full progress of the metallurgical interface reaction, thereby improving the welding strength. Under the optimum welding current of 65 A, the maximum strength of the weldment reached 219 MPa, which was 83% of the base metal of the Mg alloy. After the ultrasound treatment, the maximum tensile strength of the galvanized steel/AZ31B weldment was 251 ± 4.1 MPa. Compared with the traditional TIG welded galvanized steel/AZ31B joint, the welding strength was increased by 14.6%. In addition, compared to the application of vibration to the welding platform, the process of applying ultrasonic vibration to the welding wire in this study required less ultrasonic power. The increase of the weld mechanical properties was mainly attributed to the microstructure refinement and defect elimination, as mentioned above. Nevertheless, the mechanical properties of the weldment were slightly reduced as the ultrasonic power increased to 1.5 kW.

Figure 11 presents the stress/strain curves of the galvanized steel/AZ31B weldments obtained with and without UVAW. It is apparent that the microstructure refinement induced by UVAW not only enhanced the breaking strength but also improved the deformation ability of the galvanized steel/AZ31B joint.

Figure 12 and Figure 13 present the typical failure features of untreated and ultrasound-treated galvanized steel/Mg alloy joints. For the untreated joint, the fracture path ran through the welding zone, and the whole fracture was mainly composed of a large-scale cleavage surface and cleavage step, as shown in Figure 12a and Figure 13a. For the ultrasonic-treated joint, the fracture occurred at the edge of the fusion welding zone. It can be seen that the number of cleavage platforms of the fracture surface was fewer than that of the untreated joint and a few dimples were formed. It can be concluded that the fracture mechanism of the joint changed from a cleavage fracture to a mixed fracture of cleavage fracture and dimple fracture under the action of ultrasonic vibration.

## 4. Conclusions

It is confirmed that the sound galvanized steel/AZ31B joint was obtained by the ultrasonic vibration-assisted welding process. The main conclusions are as follows:The application of ultrasonic vibration improved the wetting of welds and eliminated porosity defects.The mechanical stirring effect caused by ultrasonic vibration made the interfacial reaction layer of the brazing zone change from wavy to laminar.Ultrasound-induced acoustic streaming and cavitation significantly refined the microstructure of the weld zone, and its average grain size was about 39 ± 1.7 µm.With ultrasonic power of 0.9 kW and a current of 65 A, the maximum tensile strength of the ultrasound-treated galvanized steel/Mg alloy joint was 251 ± 4.1 MPa, which was a 14.6% increase from the joint without ultrasonic treatment.

In this study, an effective ultrasonic vibration-assisted welding process was developed to achieve the welding of galvanized steel/AZ31B alloys. In the next step, the author will study the influences of ultrasonic vibration direction and frequency on the microstructure and mechanical properties of the joint. 

## Figures and Tables

**Figure 1 materials-14-01674-f001:**
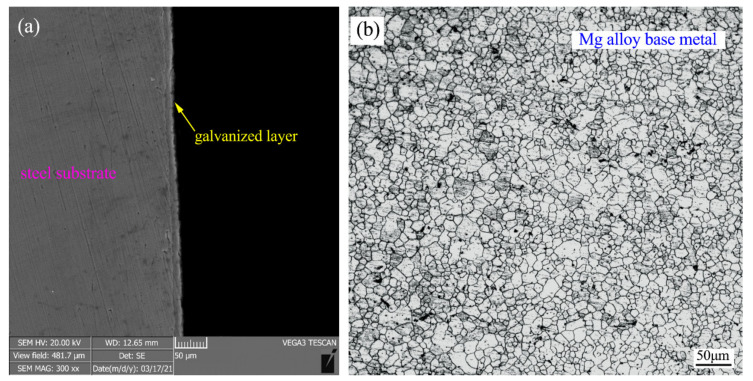
(**a**) Microstructure of galvanized steel and (**b**) Mg alloy base material.

**Figure 2 materials-14-01674-f002:**
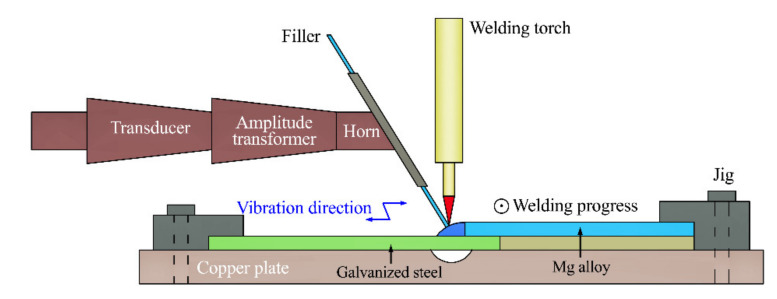
The sketch map of the ultrasonic vibration-assisted welding (UVAW) process.

**Figure 3 materials-14-01674-f003:**
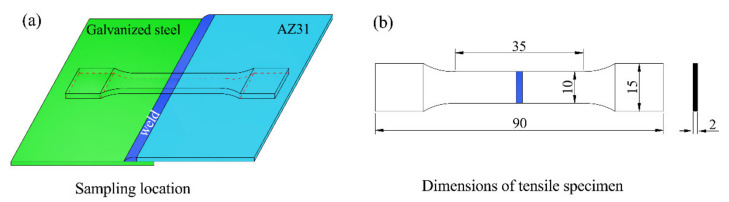
(**a**) The sampling point of tensile test samples and (**b**) the outline dimensions of tensile test samples (mm).

**Figure 4 materials-14-01674-f004:**
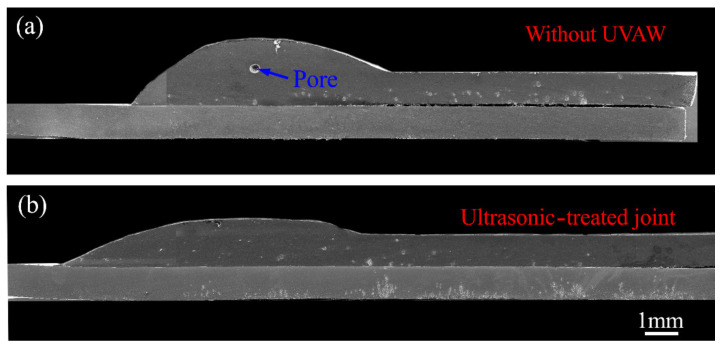
The typical cross-sectional area of weldments obtained with and without UVAW: (**a**) current of 65 A, ultrasonic power of 0 kW; (**b**) current of 65 A, ultrasonic power of 0.9 kW.

**Figure 5 materials-14-01674-f005:**
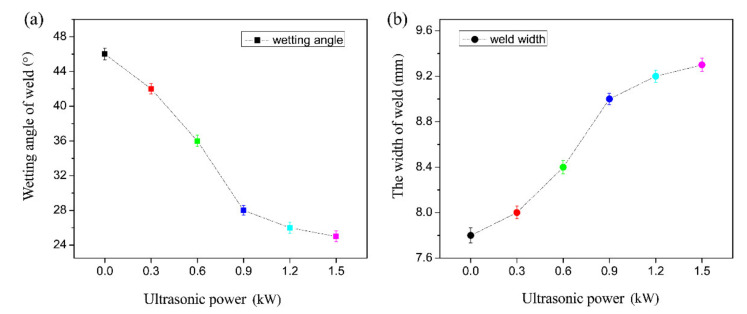
(**a**) The wetting angle and (**b**) width of weld under the assistance of varying ultrasonic power.

**Figure 6 materials-14-01674-f006:**
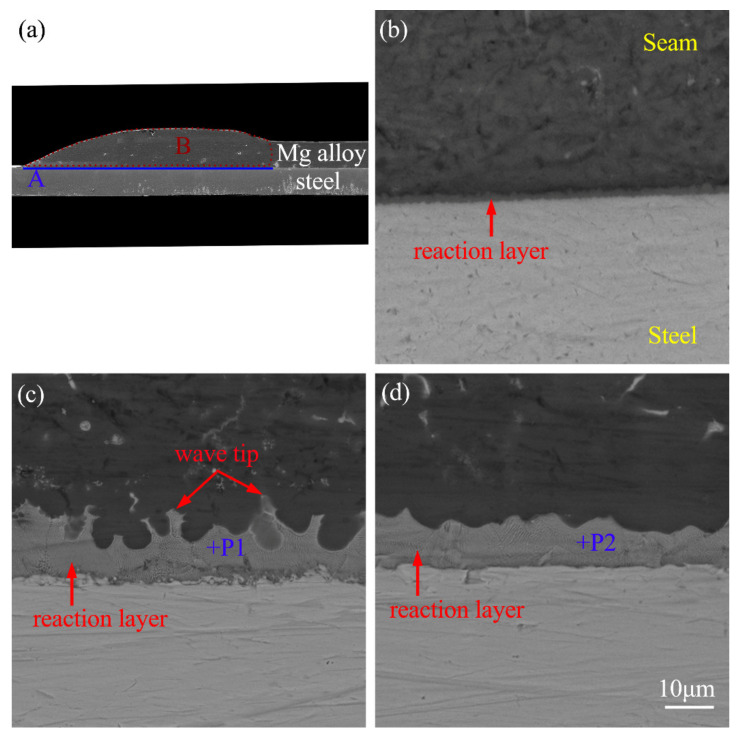
The typical backscattered electron (BSE) micrographs of the brazing area: (**a**) cross-sectional macrostructure; (**b**) current of 55 A, ultrasonic power of 0 kW; (**c**) current of 65 A, ultrasonic power of 0 kW; (**d**) current of 65 A, ultrasonic power of 0.9 kW.

**Figure 7 materials-14-01674-f007:**
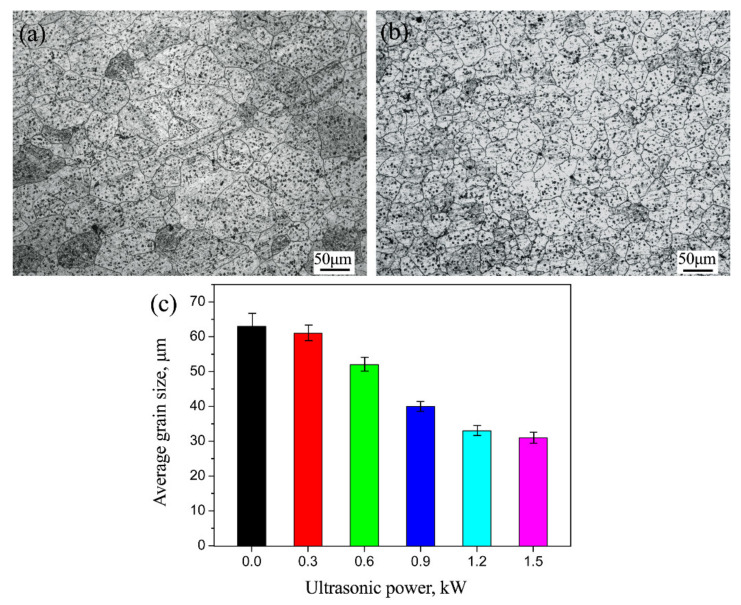
The metallographic microstructure of the fusion welding area: (**a**) ultrasonic power of 0 kW; (**b**) ultrasonic power of 0.9 kW, and (**c**) the crystallite dimension of the weld zone treated with varying ultrasonic power.

**Figure 8 materials-14-01674-f008:**
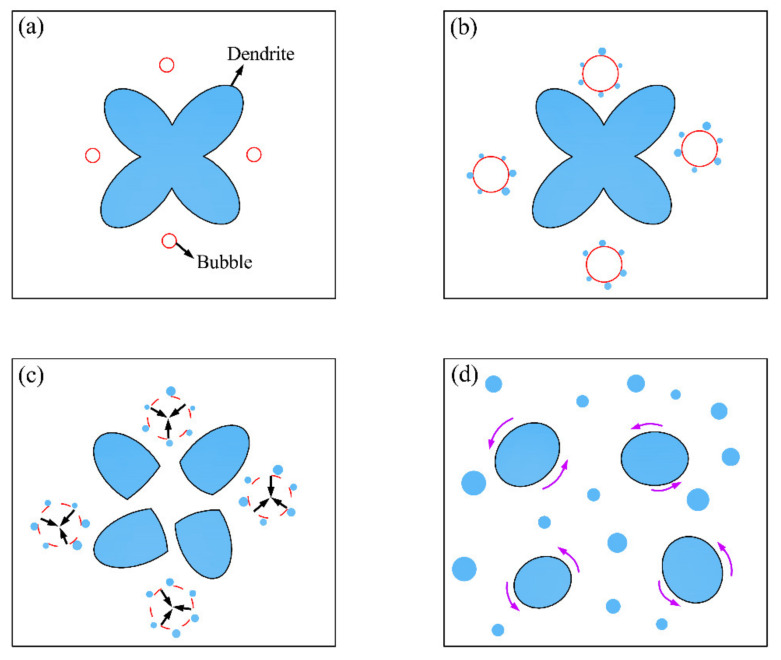
The refinement mechanism of the UVAW process: (**a**) formation of cavitation bubbles; (**b**) growth of cavitation bubbles; (**c**) the bursting of a cavitation bubble; and (**d**) effect of acoustic streaming.

**Figure 9 materials-14-01674-f009:**
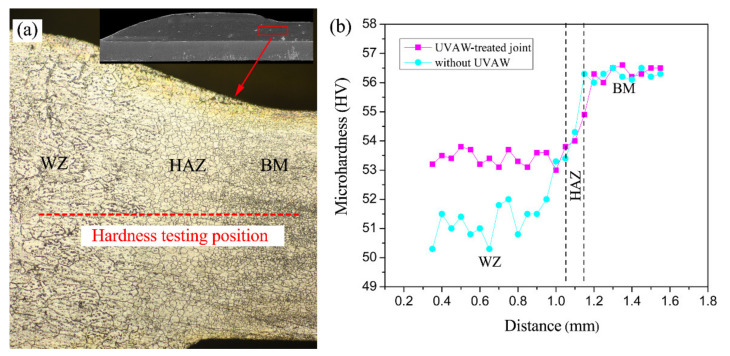
(**a**) The cross-sectional microhardness test point and (**b**) the microhardness curves of the galvanized steel/AZ31B joints obtained with and without UVAW.

**Figure 10 materials-14-01674-f010:**
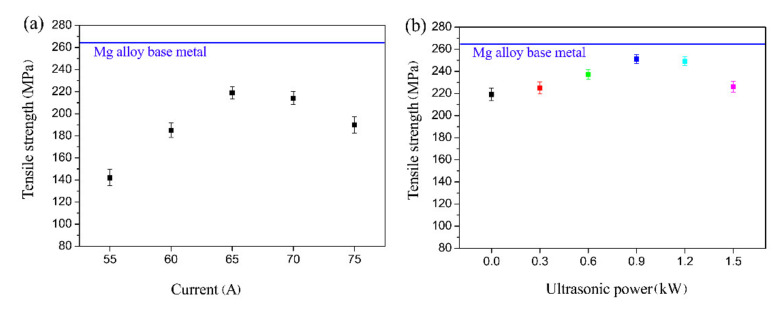
The tensile strength of the galvanized steel/AZ31B weldments: (**a**) with various currents and without UVAW; (**b**) with a current of 65 A and varying ultrasonic power.

**Figure 11 materials-14-01674-f011:**
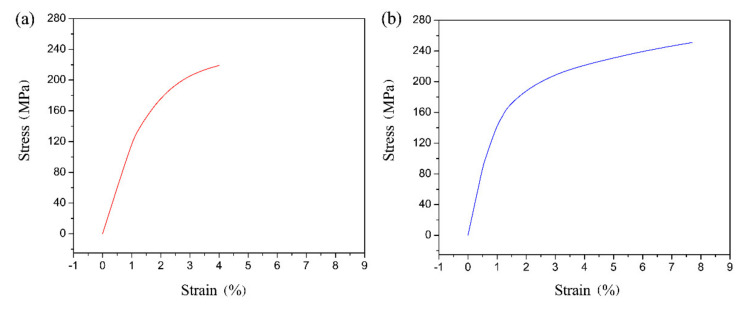
The tensile stress/strain curves of the galvanized steel/AZ31B joints: (**a**) a current of 65 A and without UVAW, (**b**) a current of 65 A and ultrasonic power of 0.9 kW.

**Figure 12 materials-14-01674-f012:**
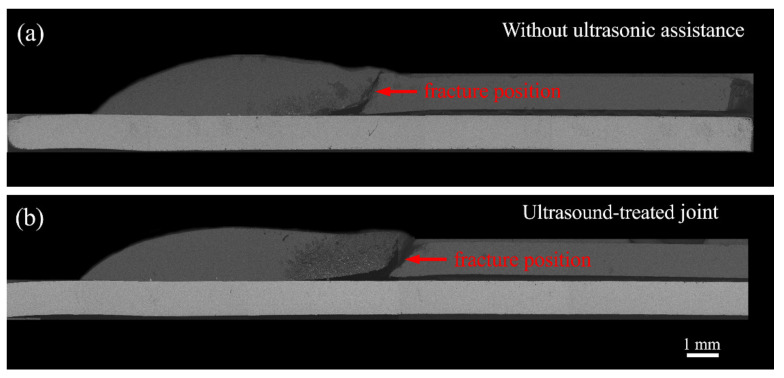
The typical fracture locations of the galvanized steel/Mg alloy joint: (**a**) a current of 65 A, ultrasonic power of 0 kW and (**b**) a current of 65 A, ultrasonic power of 0.9 kW.

**Figure 13 materials-14-01674-f013:**
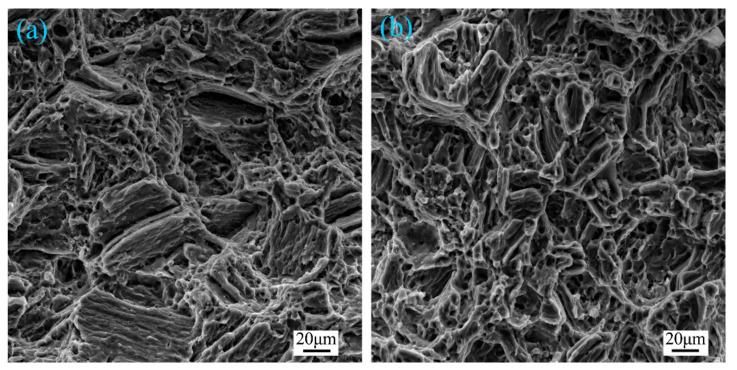
The typical fracture surfaces of the galvanized steel/AZ31B weldment: (**a**) a current of 65 A, ultrasonic power of 0 kW and (**b**) a current of 65 A, ultrasonic power of 0.9 kW.

**Table 1 materials-14-01674-t001:** The main chemical composition of the parent metals (wt.%).

Materials	Mn	Si	C	Zn	Al	Fe	Mg
Steel	0.65	0.25	0.08	-	-	Bal.	-
AZ31	0.41	0.05	-	0.88	2.90	-	Bal.
Filler	0.32	-	-	0.79	2.80	-	Bal.

**Table 2 materials-14-01674-t002:** The welding parameters used during the UVAW process.

Parameters	Value
Voltage	14 V
Wire feed speed	0.5 m/min
Welding speed	0.2 m/min
Welding current	55–75 A
Ultrasonic power	0–1.5 kW

## Data Availability

Data sharing is not applicable for this paper.

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
