# Peer review of "Microstructure Evolution and Strengthening Mechanism of Galvanized Steel/Mg Alloy Joint Obtained by Ultrasonic Vibration-Assisted Welding Process"

_materials, 2021, doi:10.3390/ma14071674_

Round 1

Reviewer 1 Report

A number of experimental details such as steel microstructure and coating thickness are missing. Some results error bar is missing. One of the major concerning things is the microstructure observation location and fracture location is completely different which does not explain the tensile properties difference between without UVAW and with UVAW. The detailed comments are attached which should be addressed.

Author Response

Dear Reviewer,

Thank you for the detailed and valuable comments on our manuscript.

We have made all necessary changes and supplements in the revised manuscript. The detailed comments are explained as follows.

Thank you!

Comments from Reviewer:

Question 1: Mention average grain size in line 13.

Response: Dear reviewer, thank you for your comments. The average grain size of fusion welding zone with error range was provided in the revised manuscript. (Line 13)

Question 2: Error bar in line 17 in tensile strength.

Response: Dear reviewer, thank you for your comments. Error bar was added in the revised manuscript. (Line 17)

Question 3: Line 77 how thick is galvanized coating?

Response: Dear reviewer, thank you for your comments. The average thickness of the galvanized layer on the steel surface was about 14±0.6μm, which was added in the revised manuscript. (Line 80)

Question 4: Parent material thickness?

Response: Dear reviewer, thank you for your comments. The parent material thickness was provided in the revised manuscript. (Line 97 and Figure 3)

Question 5: Error bar in line 82 in grain size.

Response: Dear reviewer, thank you for your comments. Error bar was added in the revised manuscript. (Line 85)

Question 6: Parent steel microstructure is required with grain size.

Response: Dear reviewer, thank you for your comments. Galvanized steel belongs to the ferritic and martensitic duplex steel system, which has an average grain size of about 17±1.1μm.

Question 7: What is welding rate in table 1? Welding speed?

Response: Dear reviewer, thank you for your comments. Welding speed was provided in the revised manuscript.

Question 8: Give EDS details in line 109.

Response: Dear reviewer, thank you for your comments. EDS details were added in the revised manuscript.

Question 9: Check the terminology in line 111 “steady velocity”. Is it strain rate?

Response: Dear reviewer, thank you for your comments. The correct terminology was provided in the revised manuscript.

Question 10: Check Figure 3 caption. What is the standard of tensile dimension? Thickness?

Response: Dear reviewer, thank you for your comments. The standard of tensile dimension and thickness were added in the revised manuscript.

Question 11: How welding zone and brazing zone was distinguished from macro image?

Response: Dear reviewer, thank you for your comments. A clearer designation will indicate the location of the welding and brazing zones in the macro image.

Question 12: Line 125-126…give reference of hydrogen trapping.

Response: Dear reviewer, thank you for your comments. Reference refer to hydrogen trapping was added in the revised manuscript.

Question 13: Line 133-134; Show the seam width and wetting angle measurement method in experimental section.

Response: Dear reviewer, thank you for your comments. The measurement method of seam width and wetting angle was provided in the revised manuscript.

Question 14: Error bar in Figure 5.

Response: Dear reviewer, thank you for your comments. Error bar was added in the revised Figure 5.

Question 15: what’s the approximate heat input?

Response: Dear reviewer, thank you for your comments. The tensile testing reveals that the sound Mg/steel joint can be obtained with the optimal welding current of 65A.

Question 16: Line 157-158: reference?

Response: Dear reviewer, thank you for your comments. The related reference was added in the revised manuscript.

Question 17: Please indicate wave like morphology and wave tip in figure.

Response: Dear reviewer, thank you for your comments. The wave tip was marked in the revised figure.

Question 18: Line 170-174“EDS was carried out and the results show that phase P1 and 170 P2 mainly contained 48.2 at. pct Zn, 51.8 at. pct Mg and 47.6 at. pct Zn, 52.4 at. pct Mg 171 respectively. It was suggested that both phase P1 and P2 were MgZn, which indicating UVAW process only changed the morphology of the reaction layer, but had no influence on the phase composition.” Does it mean iron did not melt? Line 147 “molten metal” refer to what?

Response: Dear reviewer, thank you for your comments. During the welding-brazing of Mg/steel, the galvanized layer of steel surface melt, while the steel substrate did not melt. And the Mg alloy parent plate, welding wire and galvanized layer melt to form the molten metal.

Question 19: Line 183 : error bar in grain size.

Response: Dear reviewer, thank you for your comments. The error bar was added in the revised manuscript.

Question 20: Improve the figure caption of figure 7. Figure should be self explanatory. Add error bar in figure 7c.

Response: Dear reviewer, thank you for your comments. The figure caption of figure 7 was improved to be self-explanatory. And error bar was added in figure 7c.

Question 21: The higher heat input (since ultrasonic power is increasing) should increase grain size . why it is decreasing?

Response: Dear reviewer, thank you for your comments. Fig. 8 illustrates the refinement mechanism of the UVAW process on the microstructure of the weld zone. Literatures reveal that the ultrasonic cavitation and acoustic streaming caused by ultrasonic treatment account for the refinement of coarsening grain in the fusion welding zone. As presented in Fig. 8a, a large number of cavitation bubbles were generated in the negative half cycle of the ultrasonic sine wave. Then the cavitation bubbles absorbed the heat around them and grew up rapidly, resulting in supercooling in the vicinity and thus forming a mass of nuclei, as illustrated in Fig. 8b. When the cavitation bubble grew to the threshold value, it will burst and produce a shock wave of gigapascal, thus breaking the growing dendrites, as shown in Fig. 8c. Therefore, both undercooled nucleation and broken dendrite can improve the nucleation rate and refine the microstructure of fusion zone. In addition, during the process of ultrasonic vibration propagation in the weld zone, the sound pressure gradient caused by viscosity attenuation will make the liquid metal flow. Under the action of acoustic streaming, the supercooled nucleation points and broken dendrites were evenly transported to various areas of the molten pool. In addition, acoustic flow accelerated the uniform distribution of elements and alleviated the enrichment of solute at the solidification front, which eventually promoted the spheroidization of α-Mg grains, as shown in Fig. 8d.

Question 22: Use same scale in Figure 9a&b.

Response: Dear reviewer, thank you for your comments. The same scale was applied in the revised Figure 9a and b.

Question 23: What about ductility. Add stress strain curve for comparison.

Response: Dear reviewer, thank you for your comments. The ductility was discussed in the revised manuscript.

Question 24: Figure 10 and Figure 11: “For un-ultrasonic treated joint, the fracture path ran through the welding zone” “For ultrasonic treated joint, fracture occurred at the edge of fusion welding zone”. I see big void in the interface (brazing zone) for both condition making it look like crack initiation source. Please clarify. The macro failure image showing that root of weld could be the problem considering large void.

Response: Dear reviewer, thank you for your comments. For un-ultrasonic treated joint, the fracture path ran through the welding zone, and the whole fracture was mainly composed of large-scale cleavage surface and cleavage step, as shown in Fig. 11a and Fig. 12a. For ultrasonic treated joint, fracture occurred at the edge of fusion welding zone. It can be found that the quantity of cleavage platforms of fracture surface was less than that of untreated joint and a few dimples were formed. It can be concluded that the fracture mechanism of the joint changed from cleavage fracture to mixed fracture of cleavage fracture and dimple fracture under the action of ultrasonic vibration.

Question 25: Hardness section should be before tensile tests results.

Response: Dear reviewer, thank you for your comments. The hardness section was placed before tensile tests results in the revised manuscript.

Question 26: Draw boundary in Figure12b to show the transition position. Did the failure occur in HAZ?

Response: Dear reviewer, thank you for your comments. The boundary was added in the revised manuscript. And the analysis of fracture characteristics shows that the joint failure occurs in the fusion zone of the joint.

Question 27: Microstructure observation location and fracture location is different. How it can be justified?

Response: Dear reviewer, thank you for your comments. The microstructure observation location and fracture location are located in the fusion welding zone in the revised manuscript.

Question 28: All the figures caption should be self explanatory. Every figure subsection should be explained separately.

Response: Dear reviewer, thank you for your comments. All the figure captions were carefully checked and improved to be self-explanatory.

Reviewer 2 Report

The Authors present novel technique for enhancing joining of galvanized steel and Mg alloy. In this case, Authors used novel ultrasonic vibration assisted welding (UVAW) technique. The idea of such a study is new and the manuscript does contain novel results. Based on the obtained results, a detailed description of the influence of the applied UVAW on microstructure, mechanical properties (tensile strength, microhardness) is presented for lap joints. The paper is interesting and useful. The presented analysis is logical, and the number of references is sufficient. In general, the presented results are of interest to scientists and engineers. Results of the research are relatively clear, but the manuscript needs a minor revision.  The only one minor comment is given:

Line 97. Are the welding parameters based on the own research results or based on parameters described in the literature? If the parameters were used according to existed literature database, please add a proper reference

Line 171. Reviewer recommends showing the EDS results in %weight, additionally. It will be better to compare the results in table as well.

Figure 12. Hardness investigation results is shown in Fig 12 for the weld zone horizontally. Did the Authors perform research for vertical direction, which goes through the brazing area?

Author Response

Dear Reviewer,

Thank you for the detailed and valuable comments on our manuscript.

We have made all necessary changes and supplements in the revised manuscript. The detailed comments are explained as follows.

Thank you!

Comments from Reviewer:

Question 1: Line 97. Are the welding parameters based on the own research results or based on parameters described in the literature? If the parameters were used according to existed literature database, please add a proper reference.

Response: Dear reviewer, thank you for your comments. A large number of experimental parameters have been verified by the author in the early stage. Finally, welding current and ultrasonic power were selected as the variable parameters of magnesium steel welding. Table 1 shows the welding parameters used during UVAW process.

Parameters

Value

Voltage

14 V

Wire feed speed

0.5 m/min

Welding speed

0.2 m/min

Welding current

55-75 A

Ultrasonic power

0-1.5 kW

Question 2: Reviewer recommends showing the EDS results in % weight, additionally. It will be better to compare the results in table as well..

Response: Dear reviewer, thank you for your comments. The relevant changes have been highlighted in the revised manscript.

Question 3: Hardness investigation results is shown in Fig 12 for the weld zone horizontally. Did the Authors perform research for vertical direction, which goes through the brazing area?

Response: Dear reviewer, thank you for your comments. During the ultrasonic assisted welding process of magnesium/steel, the microstructure of the fusion welding zone was improved obviously through the cavitation and acoustic flow effect induced by ultrasonic vibration, and the fracture finally occurs in the fusion welding zone. On the other hand, ultrasonic vibration did not have a significant impact on the microstructure of the brazed area of the joint, and its phase and hardness did not change significantly.

Reviewer 3 Report

The reviewer comments of the paper «Microstructure evolution and strengthening mechanism of galvanized steel/Mg alloy joint obtained by ultrasonic vibration assisted welding process»

- Reviewer

The authors presented an article «Microstructure evolution and strengthening mechanism of galvanized steel/Mg alloy joint obtained by ultrasonic vibration assisted welding process». However, there are several points in the article that require further explanation.

Comment 1:

Introduction.

It is useful to add a paragraph for industrial applications of steel/Mg allo. Show the advantages and disadvantages in comparison with the classical method of obtaining.

Group citstions, such as [1-3], [4-7], [10-14], etc. should be avoided. To do this, you can break down the sentence, for example, "Especially, the non-ferrous alloys based on magne-26 sium (Mg) [...], aluminum (Al) [..., …] and titanium (Ti) [...]".

Add an article in the introduction: Advances in Materials Science and Engineering 2019:1-12. doi:10.1155/2019/4156176

After the purpose of the article, briefly describe what has been done in each section.

Comment 2:

  1. Materials and Methods

Instead of lines 77, 78, 79, give in the table the chemical composition of the materials studied in the article.

What is the hardness of these materials? How is it measured? Add a hardness test method.

For devices and machine used in research, indicate in parentheses (manufacturer, city, country).

Comment 3:

  1. Results and Discussion

Remove the vertical line on the left of Figures 9, 12.

Are all the formulas in the article original? If not needed appropriate citations.

Comment 4:

It will be useful to add a section of Nomenclature in which to sign all the physical quantities and abbreviations encountered in the article. There are many physical quantities in the text and such a section will help to find the description of the necessary element.

For example,

UVAW  : Ultrasonic vibration assisted welding

etc.

Comment 5:

Conclusions.

It is necessary to more clearly show the novelty of the article and the advantages of the proposed method. Show practical relevance.

Comment 6:

Proofreading by a native English speaker is required.

The article is interesting. Authors should carefully study the comments and make improvements to the article step by step. All changes must be highlighted. After major changes can an article be considered for publication in the "Materials".

Author Response

Dear Reviewer,

Thank you for the detailed and valuable comments on our manuscript.

We have made all necessary changes and supplements in the revised manuscript. The detailed comments are explained as follows.

Thank you!

Comments from Reviewer:

Question 1: Group citstions, such as [1-3], [4-7], [10-14], etc. should be avoided. To do this, you can break down the sentence, for example, "Especially, the non-ferrous alloys based on magne-26 sium (Mg) [...], aluminum (Al) [..., …] and titanium (Ti) [...]". Add an article in the introduction: Advances in Materials Science and Engineering 2019:1-12. doi:10.1155/2019/4156176.

Response: Dear reviewer, thank you for your comments. The author added the above literatures in the revised draft and revised the article according to the comments of reviewers.

Question 2: Instead of lines 77, 78, 79, give in the table the chemical composition of the materials studied in the article. What is the hardness of these materials? How is it measured? Add a hardness test method. For devices and machine used in research, indicate in parentheses.

Response: Dear reviewer, thank you for your comments. The chemical composition of the base materials and filler were shown in Table 1. Hardness measurement methods and equipment details have been described in the revised draft.

Table 1. The main chemical composition of parent metal (wt.%).

Materials

Mn

Si

C

Zn

Al

Fe

Mg

steel

0.65

0.25

0.08

-

-

Bal.

-

AZ31

0.41

0.05

-

0.88

2.90

-

Bal.

Filler

0.32

-

-

0.79

2.80

-

Bal.

Question 3: Remove the vertical line on the left of Figures 9, 12. Are all the formulas in the article original? If not needed appropriate citations

Response: Dear reviewer, thank you for your comments. The vertical lines on the left of Figure 9 and 12 were removed. And the appropriate citations about formula was added in the revised manuscript.

Question 4: It will be useful to add a section of Nomenclature in which to sign all the physical quantities and abbreviations encountered in the article.

Response: Dear reviewer, thank you for your comments. All abbreviations were correctly defined as they first appeared.

Question 5: It is necessary to more clearly show the novelty of the article and the advantages of the proposed method.

Response: Dear reviewer, thank you for your comments. The novelty and advantages of the proposed method were added in the revised manuscript.

Question 6: Proofreading by a native English speaker is required.

Response: Dear reviewer, thank you for your comments. An English native associate professor has carefully modified the English expression of this article, hoping to meet the requirements of the journal.

Reviewer 4 Report

In the paper are presented an analysis of the properties of a joint made by ultrasonic welding between a galvanized steel and a magnesium alloy.

From the analysis of the information presented in the article, I found the following:

- The paper presents a series of results that could be of interest to the scientific community:

- The introduction needs to be improved. References to bibliographic sources such as 10-14 should be avoided. For example, in order to avoid such references, other bibliographic sources should be considered (Use of Ultrasound in Reconditioning by Welding of Tools Used in the Process of Regenerating Rubber, Ultrasound influence on materials structure in parts reconditioned by welding with ultrasonic field, Ultrasound effect on the mechanical properties of parts loaded by welding)

- The research methodology is not clear. The number of test pieces performed and the technological parameters used to obtain them are not presented. This is necessary because the graphical analyzes performed show property values ​​for different samples.

- The materials and methods section does not provide clear information on the equipment used in the research. Also, in this section the authors do not refer to the measurement of the hardness of the materials (this is done in the results section)

- information on the use of ultrasound is incomplete. They should be supplemented with ultrasound frequency, ultrasound amplitude, geometry of the components of the ultrasonic system (transducer, amplitude transformer, chimney)

- Attention in the text when referring to the abrasive paper appears the character # !!!!!;

- The points where the hardness measurement is performed must be presented in more detail;

- I could not identify the scientific novelty brought by the results presented in the paper. In this sense, the discussion part must be completed with some information that will show the novelty brought by the research in the paper compared to other research in the field.

- In the final part of the conclusions, the future research directions must be presented.

Author Response

Dear Reviewer,

Thank you for the detailed and valuable comments on our manuscript.

We have made all necessary changes and supplements in the revised manuscript. The detailed comments are explained as follows.

Thank you!

Comments from Reviewer:

Question 1: The introduction needs to be improved. References to bibliographic sources such as 10-14 should be avoided.

Response: Dear reviewer, thank you for your comments. The author rewrote the introduction part and quoted the references correctly in the revised manuscript.

Question 2: The research methodology is not clear. The number of test pieces performed and the technological parameters used to obtain them are not presented.

Response: Dear reviewer, thank you for your comments. The author added detailed experimental details in the revised draft.

Question 3: The materials and methods section does not provide clear information on the equipment used in the research. Also, in this section the authors do not refer to the measurement of the hardness of the materials (this is done in the results section).

Response: Dear reviewer, thank you for your comments. Detailed equipment information and hardness measurement methods were provided in the revised manuscript.

Question 4: information on the use of ultrasound is incomplete. They should be supplemented with ultrasound frequency, ultrasound amplitude, geometry of the components of the ultrasonic system (transducer, amplitude transformer, chimney)

Response: Dear reviewer, thank you for your comments. A set of self-made ultrasonic vibration auxiliary welding system, including argon tungsten arc welding machine (YC-300WP5HGN), ultrasonic generator (GDZ-1012M), horn, transducer (DW-20k), etc., was used for the welding of galvanized steel and Mg alloy. The ultrasonic frequency is 20kHz and the amplitude is 30 μm.

Question 5: Attention in the text when referring to the abrasive paper appears the character #.

Response: Dear reviewer, thank you for your comments. The author has revised the relevant contents according to the comments of reviewers.

Question 6: The points where the hardness measurement is performed must be presented in more detail.

Response: Dear reviewer, thank you for your comments. The microhardness distributions of weld joint were evaluated along the horizontal directions with 50g load and 10s holding time (MH-5L).

Question 7: In the final part of the conclusions, the future research directions must be presented.

Response: Dear reviewer, thank you for your comments. The future research directions was presented in the revised manuscript.

Round 2

Reviewer 1 Report

Thank you for addressing a number of comments. However, adding a few more details could be very helpful

1. Adding the steel microstructure with a coating layer will be helpful for reader. Since the coating is an important context of the current manuscript

2. The author mentioned that the ductility results have been discussed. However, I don't see the discussion in the updated draft.

3.  some figure is still not self-explanatory. For example figure 8

Reviewer 3 Report

The authors have improved the article according to the comments. The article can now be published.

Author Response

Dear Reviewer,

Thank you for the comments on our manuscript. Besides, in order to improve the quality of the paper, the author made further modifications to the language expression.

Reviewer 4 Report

The introduction has not been improved adequate; for example, bibliographic source no. 7 is not related to the statement in the text; also, the text introduced between lines 34-37 is not related to the subject of the paper. Other bibliographic sources in the field can be discussed in the introductory part.

Elements related to the improvement of the research methodology were not added in the text;

The geometry of the components of the ultrasonic system (transducer, transformer, etc.) was not specified in the paper;

The points at which hardness was measured cannot be identified in the text; a figure should be added where these can be observed;

The discussions were not completed with elements that highlight the novelty brought by the research in relation to other researches in the field.

Future research directions cannot be identified in the text; these must be presented in the Conclusions section.

Author Response

Dear Reviewer,

Thank you for the detailed and valuable comments on our manuscript.

We have made all necessary changes and supplements in the revised manuscript. The detailed comments are explained as follows.

Thank you!

Comments from Reviewer:

Question 1: The introduction has not been improved adequate; for example, bibliographic source no. 7 is not related to the statement in the text; also, the text introduced between lines 34-37 is not related to the subject of the paper. Other bibliographic sources in the field can be discussed in the introductory part.

Response: Dear reviewer, thank you for your comments. Reference [7] was revised and references not related to the subject of this paper were removed. Besides, more references about ultrasonic assisted technology was added in the introduction part.

Question 2: Elements related to the improvement of the research methodology were not added in the text.

Response: Dear reviewer, thank you for your comments. The author added more experimental details in the revised manuscript.

Question 3: The geometry of the components of the ultrasonic system (transducer, transformer, etc.) was not specified in the paper.

Response: Dear reviewer, thank you for your comments. The geometry of the components of the ultrasonic system were provided in the revised manuscript.

Question 4: The points at which hardness was measured cannot be identified in the text; a figure should be added where these can be observed.

Response: Dear reviewer, thank you for your comments. A figure was added to reveal the measurement points of micro hardness testing, as shown in Fig. 9.

Question 5: The discussions were not completed with elements that highlight the novelty brought by the research in relation to other researches in the field.

Response: Dear reviewer, thank you for your comments. The author has revised the relevant contents according to the comments of reviewers.

Question 6: Future research directions cannot be identified in the text; these must be presented in the Conclusions section.

Response: Dear reviewer, thank you for your comments. The future research directions was presented in the revised conclusions section.